# Optimizing the Mulching Pattern and Nitrogen Application Rate to Improve Maize Photosynthetic Capacity, Yield, and Nitrogen Fertilizer Utilization Efficiency

**DOI:** 10.3390/plants13091241

**Published:** 2024-04-30

**Authors:** Hengjia Zhang, Tao Chen, Shouchao Yu, Chenli Zhou, Anguo Teng, Lian Lei, Fuqiang Li

**Affiliations:** 1College of Agronomy and Agricultural Engineering, Liaocheng University, Liaocheng 252059, China; gsauct@163.com (T.C.); ysc@lcu.edu.cn (S.Y.); zhouchenli2021@126.com (C.Z.); 2Yimin Irrigation Experimental Station, Zhangye 734500, China; gsmltag@163.com (A.T.); ll4426072@163.com (L.L.); 3College of Water Conservancy and Hydropower Engineering, Gansu Agricultural University, Lanzhou 730070, China; lifuq@gsau.edu.cn

**Keywords:** film mulching, nitrogen, maize, yield, nitrogen use efficiency, soil quality

## Abstract

Residual film pollution and excessive nitrogen fertilizer have become limiting factors for agricultural development. To investigate the feasibility of replacing conventional plastic film with biodegradable plastic film in cold and arid environments under nitrogen application conditions, field experiments were conducted from 2021 to 2022 with plastic film covering (including degradable plastic film (D) and ordinary plastic film (P)) combined with nitrogen fertilizer 0 (N0), 160 (N1), 320 (N2), and 480 (N3) kg·ha^−1^. The results showed no significant difference (*p* > 0.05) in dry matter accumulation, photosynthetic gas exchange parameters, soil enzyme activity, or yield of spring maize under degradable plastic film cover compared to ordinary plastic film cover. Nitrogen fertilizer is the main factor limiting the growth of spring maize. The above-ground and root biomass showed a trend of increasing and then decreasing with the increase in nitrogen application level. Increasing nitrogen fertilizer can also improve the photosynthetic gas exchange parameters of leaves, maintain soil enzyme activity, and reduce soil pH. Under the nitrogen application level of N2, the yield of degradable plastic film and ordinary plastic film coverage increased by 3.74~42.50% and 2.05~40.02%, respectively. At the same time, it can also improve water use efficiency and irrigation water use efficiency, but it will reduce nitrogen fertilizer partial productivity and nitrogen fertilizer agronomic use efficiency. Using multiple indicators to evaluate the effect of plastic film mulching combined with nitrogen fertilizer on the comprehensive growth of spring maize, it was found that the DN2 treatment had the best complete growth of maize, which was the best model for achieving stable yield and income increase and green development of spring maize in cold and cool irrigation areas.

## 1. Introduction

As one of the C4 crops with the most extensive planting area in the world, maize is a crop for food and feed and an essential source of industrial raw materials [1]. China has become the second largest producer of maize, with the planting area accounting for more than 30% of the national grain crop planting area, reaching over 40 million hm^2^ [2]. (https://www.stats.gov.cn/sj/. accessed on 25 March 2024). Thus, increasing maize yield is essential to ensuring food security, achieving self-sufficiency in food supply, and stabilizing economic development. However, the frequent occurrence of extreme drought and the shrinking of arable land area has brought enormous pressure on agricultural production and even caused decreases in food production [3,4]. How to alleviate the pressure of reduced grain production and implement a food security strategy is a major challenge currently faced.

Plastic film mulching is a key technology to improve crop yield [5] and change the agricultural production mode in areas with water shortages [6]. Plastic film mulching may improve soil moisture and heat status, promote the decomposition and transformation of soil organic matter, improve soil nutrient content, enzyme activity, and microbial richness, inhibit weed growth, reduce nutrient competition, and create a good soil environment for crop growth [7]. Plastic film covering not only increases crop yield but also hinders gas exchange between soil and atmosphere, enhances crop root respiration intensity, and strengthens soil nitrification and denitrification processes to increase N_2_O and CH_4_ gas emission pressure [8]. Ordinary film used in mulching is mainly made of polyethylene, which is degraded very slowly in soil [9,10]. However, long-term plastic film mulching and a lack of effective recovery measures make the residue of farmland mulching plastic film a critical environmental problem. Film residue may destroy the continuity of soil pores [11], change the composition of the soil microbial community [12], affect water [13] and nutrient migration [14], hinder seed germination [15] and crop root development, and ultimately reduce crop yield [16], even threatening food security. To solve the adverse effects caused by continuous plastic film covering for many years, it is essential to develop and apply new covering materials, such as degradable plastic film.

As a new plastic film, the degradable type could be degraded into CO_2_ and H_2_O with the aid of the soil and natural environments [17], which could alleviate the pressure on the agricultural ecological environment and was considered an effective way to solve the problem of farmland residual film concentration. Therefore, the degradable plastic film has been applied in many countries, such as China [18], Italy [19], Thailand [20], etc. In addition, the soil water and heat preservation effects of degradable film are equivalent to those of ordinary film [21], which can effectively improve the soil moisture and heat status [22]. In the middle and late stages of crop growth, the soil moisture and heat of the degradable plastic film were lower than those of the ordinary due to the expansion of the degradation area [23]. The difference in crop yield and water use efficiency was slight between the degradable film mulching and the ordinary [1]. Also, soil nitrate nitrogen accumulation could even be reduced under degradable film mulching [24]. At present, nitrogen fertilizer application is widespread in agricultural production to maintain high yields. However, excessive application of nitrogen fertilizer does not significantly improve crop yield and may result in yield reduction; furthermore, surplus nitrogen might be discharged into the atmosphere in gaseous form, causing environmental pollution [25]. In addition, the low nitrogen use efficiency and decline in recovery rate would lead to more soil residual nitrogen or nitrogen leaching [26], resulting in soil salinization and groundwater pollution [27,28]. Therefore, optimizing nitrogen application rate and improving nitrogen utilization are of great significance for improving grain quality, efficiency, and environmental protection.

Film mulching combined with nitrogen fertilizer application is an essential measure in agricultural production, which can significantly improve crop yield and water use efficiency, increase soil microbial nitrogen content and particulate organic nitrogen, and improve soil fertility, conducive to sustainable development of the agricultural system. Therefore, the objectives of this study were to determine: (1) effects of ordinary and degradable plastic film on dry matter accumulation and physiological aspects of spring maize under different nitrogen application gradients; (2) performance in crop yield, water and nitrogen use efficiency, and soil quality subjected to nitrogen application; and (3) the possibility of degradable plastic film replacing the ordinary by multiple indicators.

## 2. Materials and Methods

### 2.1. Description of the Study Site

The trial was conducted at the Yimin irrigation experimental station in Minle County, Gansu Province, China, from April 2021 to October 2022. The area is located at 100°43′ east longitude, 38°39′ north latitude, and 1970 m above sea level, belonging to a temperate continental climate (Figure 1). The average annual precipitation is about 200 mm, with the evaporation of 1680–2270 mm; the average sunshine duration is about 2592–2997 h; the average yearly temperature is 3.4–5.6 °C; and the frost-free period is about 78–188 d. The tested soil was light loam, with a maximum field water capacity of 24% and a soil bulk density of 1.46 g·cm^−3^ in topsoil. The soil pH is 7.2 and soil fertility is medium within 0–20 cm soil layer, with organic matter of 12.6 g·kg^−1^ and the available phosphorus, potassium, and alkali hydrolyzed nitrogen of 15.8 mg·g^−1^, 192.1 mg·kg^−1^ and 57.5 mg·kg^−1^, respectively. The rainfall in 2021 and 2022 was 244.7 mm and 237.4 mm, respectively (Figure 2).

### 2.2. Experimental Design and Field Management

The spring maize crop was film-mulched and fertilized with nitrogen. There were two kinds of mulching film: the ordinary mulching film and the egradable mulching film (produced by Shandong Tianzhuang environmental protection Co., Ltd. with a thickness of 0.008 mm and a width of 70 cm, Jinan, China), respectively recorded as P, D. There were four nitrogen application levels: 0, 160, 320, and 480 kg·ha^−1^, respectively, recorded as N0, N1, N2, and N3. There were 8 treatments in total, with 3 replications. There were 18 plots with each area of 28 m^2^ (2 m × 14 m), a 0.2 m interval between communities. The field plots were arranged with random blocks. The spring maize was planted with a row spacing of 40 and plant spacing of 35cm, and a planting density of 74,000 plants per hectare, sown on 16 April and harvested on 25 September 2021 or sown on 18 April and harvested on 28 September 2022, respectively. The fields were rotary tilled and leveled before sowing. The application amounts of phosphorus fertilizer (P_2_O_5_) and potassium fertilizer (K_2_O) were the same at 120 kg·ha^−1^ and 80 kg·ha^−1^, respectively, and all fertilizers were supplied as base fertilizer, which was applied to the soil when the soil was turned. Nitrogen fertilizer was applied four times at different growth stages, namely, 20% as base fertilizer, 30% at jointing, 30% at tasseling, and 20% at grain filling. The crop was watered using plastic film-mulched drip irrigation with the same irrigation amount according to 100%ETc (ETc = Kc × ET0, ET0 is calculated based on Penman–Monteith equation recommended by FAO, while Kc refers to the standard of the China Meteorological Administration; the Kc values for April, May, June, July, August, and September are 0.3, 0.4, 0.88, 1.26, 1.25, and 0.73, respectively) [29] (https://hbba.sacinfo.org.cn/, accessed on 10 January 2024). The meteorological parameters were provided by the micro-meteorological instrument system in the experimental station. The effective rainfall in the 2021 and 2022 crop growing seasons was 138.86 mm and 123.66 mm, respectively (Figure 3), and the irrigation amount in the above two growing seasons was 627 mm and 609 mm, respectively.

### 2.3. Measurements and Calculations

#### 2.3.1. Above-Ground and Underground Biomass

Three plants were randomly selected with uniform jointing, tasseling, and grain filling of spring maize. The root sampling area was 15 × 15 cm around the plant, and the sampling depth was determined according to the depth of spring maize roots. The plants were decomposed into different organs, then killed at 105 °C for 30 min, and finally dried at 80 °C to a constant weight. The dry weight of each organ was weighed, and the root/shoot ratio was calculated according to equation R/S(%) = root biomass/above-ground biomass.

#### 2.3.2. Photosynthetic Gas Exchange Characteristics

Photosynthetic gas exchange parameters at the third leaf of the spring maize ear with three repetitions were measured at 9:00–11:00 a.m. on sunny days during spring maize jointing, tasseling, and grain filling using a LI-6400 portable photosynthesis instrument, including photosynthetic rate, stomatal conductance, and transpiration rate.

#### 2.3.3. Soil Quality

During the maize harvest period, 0–20 cm of soil was taken in the middle of two corn plants, and 3 points were randomly taken from each treatment as mixed samples to measure soil enzyme activity. The urease was measured by sodium phenol and sodium hypochlorite colorimetry. The enzyme activity was expressed by the milligrams of NH^3^-N produced by 1 g of soil after incubation at 37 °C for 24 h under the action of urease. The sucrase was measured using the 3,5-dinitrosalicylic acid colorimetric method, and enzyme activity was expressed as the milligrams of glucose produced in 1 g of soil after being incubated at 37 °C for 24 h under the action of sucrase. The soil pH was measured by a pH meter (PHS-3C), with a soil mass extract of 2.5:1.

#### 2.3.4. Grain Yield and Its Components

Ten spring maize plants were randomly selected in each plot at spring maize ripening to determine the grain yield after measuring the yield components, including grain number per ear, row number per ear, ear longitudinal diameter, and ear diameter.

#### 2.3.5. Water and Nitrogen Use Efficiency

The crop evapotranspiration (ET, mm) was calculated using the following equation [30]:(1)ET=P+I+U−D−S+ΔW
where P is the effective precipitation (mm); I is the amount of irrigation (mm); U is the amount of groundwater recharge (mm). The depth of groundwater is below 20 m, so groundwater recharge can be ignored. D is the amount of deep leakage (mm) (the tested area is flat, thus there is no surface runoff, therefore D = 0). ΔW is the soil water storage change between plant sowing and harvest (mm).

The water use efficiency (WUE, kg·m^−3^) was calculated according to following formula:(2)WUE=Y/ET
where Y is spring maize grain yield (kg·ha^−1^).

The irrigation water use efficiency (IWUE, kg·m^−3^) was calculated using the following formula:(3)IWUE=Y/I

The nitrogen fertilizer partial productivity (NPF, kg·kg^−1^) was calculated according to the following formula:(4)$$NPF=Y_{N}/N$$
where Y_N_ is the spring maize yield in nitrogen application area (kg·ha^−1^), and N is the amount of nitrogen fertilizer input (kg·ha^−1^).

The nitrogen fertilizer agronomic use efficiency (NFA, kg·kg^−1^) was calculated using the following formula:(5)$$NFA=\left(Y_{N}-Y_{0}\right)Y_{N}/N$$
where Y_0_ is the spring maize yield in the area without nitrogen application (kg·ha^−1^).

### 2.4. Statistical Analysis

The SPSS 22.0 software was used to analyze the difference in the measured data (*p* < 0.05), and the Origin 2021 software was used for plotting. The Yaaph software (http://www.jeffzhang.cn/, accessed on 25 March 2024) was used to draw the comprehensive analysis hierarchy model of spring maize and the weight analysis of each index. The Matlab software (https://ww2.mathworks.cn/products/matlab.html, accessed on 25 March 2024) was used to calculate the weight of the combination based on the game theory and the comprehensive score of TOPSIS.

## 3. Result

### 3.1. Root and Shoot Growth

Nitrogen fertilizer is the main factor affecting spring maize root and shoot growth. The degradable plastic film was gradually degraded with spring maize growth, and the effects of different film types on spring maize growth were quite different, showing significant (*p* < 0.05) effects on spring maize growth at tasseling and grain filling and significant (*p* < 0.01) effects on spring maize root and shoot growth (Table 1).

#### 3.1.1. Above-Ground Dry Matter

The spring maize above-ground dry matter accumulation showed an increasing trend with plant growth (Figure 4). At jointing, there was no significant difference (*p* > 0.05) between the degradable plastic film mulching and the ordinary above-ground dry matter accumulation. The above-ground dry matter was significantly improved under nitrogen application, and that in N2 treatment marked the maximum with 7.76~31.43% increase under the degradable plastic film mulching and 6.50~28.85% increase under the ordinary mulching. At tasseling, nitrogen fertilizer was the main factor affecting spring maize above-ground dry matter. Compared with N0, N1, and N3, N2 treatment, the above-ground dry matter of spring maize was increased by 49.89%, 22.36%, and 7.12% under the degradable plastic film mulching and 40.39%, 20.45%, and 5.27% under the ordinary mulching, respectively. At the grain-filling stage, nitrogen fertilizer had a more significant effect on increasing above-ground dry matter accumulation. Compared with N0 and N1, N2 significantly increased by 61.45%, 28.66% under degradable plastic film mulching and 52.87%, 28.74% under ordinary mulching, and the effect of nitrogen fertilizer under degradable plastic film mulching was better than that of ordinary plastic film mulching. It can be seen that a reasonable amount of nitrogen fertilizer can promote the growth and development of spring maize and improve the dry matter quality of above-ground parts. When the nitrogen fertilizer level exceeds N2, it will inhibit the growth of spring maize and affect the accumulation of dry matter. There was no significant difference in the development of spring maize under ordinary plastic film mulching and degradable plastic film mulching (*p* > 0.05).

#### 3.1.2. Root Dry Matter

The root biomass of spring maize reached its maximum, with the growth stage advancing to the filling phase (Figure 4). Nitrogen fertilizer can promote the growth of the spring maize root system and improve its quality. At the jointing stage, N2 treatment increased the root system by 40.57%, 20.58%, and 8.07% under the degradable plastic film mulching and 37.97%, 20.50%, and 7.66% under the ordinary mulching, respectively, compared with N0, N1, and N3, indicating that N3 nitrogen application can inhibit the growth of the spring maize root system. At the tasseling stage, under the cover of degradable plastic film and ordinary plastic film, the growth rate from N0 to N1 increased by 40.39% and 31.06%, while from N1 to N2 it increased by 26.38% and 25.43%. It can be seen that the effect of root mass growth gradually decreased with the increase in nitrogen application level, and even the N3 treatment of degradable plastic film and ordinary plastic film decreased by 6.58% and 8.92%, respectively. At the grain filling stage, compared with N0, N1, and N3, N2 treatment significantly increased 115.73%, 58.17%, and 22.02% under the degradable plastic film mulching and 103.56%, 49.18%, and 18.53% under the ordinary mulching (*p* < 0.05). From the jointing stage to the grain filling period, the degradable mulching film improved root quality more than the ordinary one.

#### 3.1.3. Root Shoot Ratio

At the jointing stage, film mulching type, nitrogen application level, and their interaction had no significant effect on root shoot ratio (*p* > 0.05). From the tasseling stage to the grain filling stage, the impact of nitrogen fertilizer on the root shoot ratio reached *p* < 0.01 level, and film mulching and its interaction had no significant effect on the root shoot ratio (*p* > 0.05) (Table 2).

The root shoot ratio increased first and then decreased with the growth period (Figure 5). At a jointing stage, film mulching type and nitrogen application level had no significant effect on the root shoot ratio (*p* > 0.05). At the tasseling stage, nitrogen fertilizer could significantly improve the root shoot ratio of spring maize. The root shoot ratio of spring maize under degradable plastic film mulching increased with the increase in nitrogen application level; from N0 to N1 increased by 14.59%, from N2 to N2 increased by 3.35%, and from N2 to N3 increased by 0.15%; Under ordinary plastic film mulching, the root shoot ratio of the N3 treatment was 4.37% lower than that of N2, indicating that degradable mulching was more conducive to the growth of the spring maize root shoot and coordinated the root shoot ratio. There was no significant difference in the root shoot ratio of the nitrogen application treatment (*p* > 0.05). At the grain-filling stage, the nitrogen application level of N2 was significantly higher than that of N0 and N1 by 33.35%, 22.56% under the degradable plastic film mulching, and 32.54%, 15.69% under the ordinary mulching (*p* < 0.05), respectively. At the same time, there was no significant difference between N3 and N2 (*p* > 0.05).

### 3.2. Photosynthetic Gas Exchange Characteristics

Nitrogen fertilizer was the main factor affecting the net photosynthetic rate, transpiration rate, and stomatal conductance of spring maize, reaching a level of *p* < 0.01. Film mulching and its interaction at the jointing stage did not significantly affect the net photosynthetic rate, transpiration rate, or stomatal conductance. Nitrogen fertilizer from the tasseling location to the grain filling stage had significant (*p* < 0.05) and highly significant (*p* < 0.01) effects on net photosynthetic rate, transpiration rate, and stomatal conductance, and the interaction between film mulching and nitrogen fertilizer had no significant impact (*p* > 0.05) (Table 3).

#### 3.2.1. Net Photosynthetic Rate

With the advance of the spring maize growth period, the net photosynthetic rate reached its maximum at the tasseling stage and slightly decreased at the grain filling stage. The net photosynthetic rate increased with the increase in nitrogen application level (Figure 6). At jointing, there was no significant difference in net photosynthetic rate between N2 and N3 treatments under degradable plastic film mulching (*p* > 0.05), which was significantly increased by 19.40%, 8.95% and 24.25%, 13.37% compared with N0 and N1, respectively. Under ordinary plastic film mulching, N3 was increased by 18.90%, 10.54%, and 3.88% compared with N2, N1, and N0, respectively. At tasseling, the net photosynthetic rate of degradable plastic film and ordinary plastic film mulching increased by 16.57% and 11.45% from N0 to N1, increased by 9.24% and 7.78% from N1 to N2, and increased by 2.87% and 2.17% from N2 to N3, respectively. It can be seen that the effect of nitrogen fertilizer gradually weakened with the increase in nitrogen application level. At the grain filling stage, the nitrogen application level of N3 was significantly higher than that of N0, N1, and N2 (*p* < 0.05), increasing by 45.71%, 26.64%, and 15.10% under the degradable plastic film mulching, and 43.44%, 29.41%, and 13.32% under the ordinary mulching (*p* < 0.05), respectively. From the jointing to the grain filling stage, there was no significant difference in the net photosynthetic rate between degradable plastic film and ordinary plastic film under the same nitrogen application level (*p* > 0.05), and the increase in the net photosynthetic rate of degradable plastic film combined with nitrogen fertilizer was higher than that of ordinary plastic film.

#### 3.2.2. Transpiration Rate

The transpiration rate of spring maize increased first, then decreased with the advance of the growth period, and increased with the increase in nitrogen application level (Figure 7). At a jointing stage, the degradable plastic film and common plastic film increased by 23.20% and 10.96%, respectively, from N0 to N1, increased by 14.29% and 12.63%, respectively, from N1 to N2, and increased by 5.23% and 4.62%, respectively, from N2 to N3. This showed that nitrogen application significantly increased the transpiration rate. At the tasseling stage, the transpiration rate of nitrogen application treatment was significantly higher than that of no nitrogen application treatment (*p* < 0.05). N1, N2, and N3 under degradable plastic film cover increased by 29.50%, 54.07%, and 68.84% compared to N0, respectively. N1, N2, and N3 under ordinary plastic film cover increased by 22.81%, 40.40%, and 52.66% compared to N0. Moreover, there was no significant difference in transpiration rate between degradable plastic film cover and ordinary plastic film cover under the same nitrogen application level (*p* > 0.05). At the grain filling stage, the nitrogen application level treatment of N3 was 69.97%, 21.08%, and 5.06% under the degradable plastic film mulching, and 55.42%, 15.13%, and 8.15% under the ordinary mulching, higher than that of N0, N1, and N2, respectively. From the jointing to the tasseling stage, the transpiration rate of common plastic film mulching was higher than that of degradable plastic film mulching, but there was no significant difference (*p* > 0.05).

#### 3.2.3. Stomatal Conductance

As the growth period progressed, the stomatal conductance of spring maize reached its maximum at the tasseling stage and slightly decreased during the filling phase (Figure 8). During the jointing stage, the stomatal conductance of nitrogen application treatments was significantly higher than that of non-nitrogen application treatments (*p* < 0.05). N3, N2, and N1 increased by 11.87%, 23.50%, and 32.00% under the degradable plastic film mulching, and 11.57%, 16.67%, and 24.07% under the ordinary mulching, respectively, compared to N0. Moreover, the stomatal conductance of ordinary plastic film was higher than that of degradable plastic film, but there was no significant difference (*p* > 0.05). During the tasseling period, N3 treatment under degradable plastic film coverage was significantly higher than N2, N1, and N0 (*p* < 0.05), with increases of 48.95%, 23.48%, and 12.70%, respectively. There was no significant difference in nitrogen application levels between N3 and N2 under ordinary plastic film coverage (*p* > 0.05), both of which were significantly higher than N1 and N0 (*p* < 0.05). During the grain-filling period, under the cover of degradable and ordinary plastic film, the increase from N0 to N1 was 12.83% and 16.74%, respectively. The increase from N1 to N2 was 29.02% and 35.48%, and the increase from N2 to N3 was 17.63% and 9.79%, respectively. Under the same nitrogen application level, the stomatal conductance of ordinary plastic film was higher than that of degradable plastic film, and when the nitrogen application level was lower than N2, the amplification effect of average plastic film was better than that of degradable plastic film. Under the nitrogen application level of N3, the amplification effect of degradable plastic film was better than that of ordinary plastic film.

### 3.3. Yield and Water and Nitrogen Use Efficiency

Nitrogen fertilizer was the main factor affecting spring maize yield, water consumption, and water and nitrogen utilization efficiency, reaching *p* < 0.05 and *p* < 0.01. Film mulching has a significant (*p* < 0.05) and highly effective (*p* < 0.01) impact on irrigation water use efficiency (IWUE) and nitrogen fertilizer agronomic utilization efficiency. The interaction between film mulching and nitrogen fertilizer had a significant (*p* < 0.01) impact on nitrogen fertilizer agronomic utilization efficiency (Table 4).

Nitrogen fertilizer significantly increased the spring maize yield. Under biodegradable plastic film and ordinary plastic film coverage, the yield increased from N0 to N1 by 20.42% and 18.77%, respectively, and from N1 to N2 by 18.34% and 17.89%, respectively. As the nitrogen application rate increased, the yield of spring maize gradually weakened and even decreased by 2.01% to 3.61% at the N3 nitrogen application level. Moreover, under the same nitrogen application level, the yield of biodegradable plastic film was not significantly different from that of ordinary plastic film (*p* > 0.05). Nitrogen fertilizer promoted spring maize water absorption and increased water consumption. N3 is 8.22%, 4.65%, and 1.02% higher under the degradable plastic film mulching than N2, N1, and N0, respectively, and 8.88%, 5.04%, and 1.16% under the ordinary mulching. Increasing nitrogen fertilizer application can improve spring maize WUE and IWUE. Under degradable plastic film coverage, N2 treatment increased WUE and IWUE by 33.55%, 14.53%, 42.55%, and 18.40% compared to N1 and N0, respectively. Under ordinary plastic film coverage, N2 treatment increased WUE and IWUE by 30.08%, 13.48%, 40.09%, and 17.94% compared to N1 and N0, respectively. However, it reduced nitrogen fertilizer productivity. Degradable plastic film treatment reduced N2 and N3 by 69.01% and 163.00%, respectively, while ordinary plastic film treatment reduced N2 and N3 by 69.65% and 159.69%, respectively, compared to N1.

### 3.4. Soil Enzyme Activity and pH

Nitrogen fertilizer significantly affected the soil urease and sucrase (*p* < 0.01), while film mulching had a critical (*p* < 0.05) and highly effective (*p* < 0.01) effect on urease and sucrase. Film mulching and nitrogen fertilizer had no significant effect on pH (*p* > 0.05), and the interaction had no considerable impact on urease, sucrase, and pH (*p* > 0.05) (Table 5).

The soil urease, sucrase, and pH under plastic film mulching were lower than those under ordinary plastic film mulching, but there was no significant difference (*p* > 0.05) (Figure 9). Under ordinary plastic film coverage, urease and sucrase increased by 34.24% and 13.67% from N0 to N1, 26.72% and 15.24% from N1 to N2, and 11.18% and 1.44% from N2 to N3, respectively. Under biodegradable plastic film coverage, urease and sucrase increased by 42.21% and 21.92% from N0 to N1, 32.88% and 14.48% from N1 to N2, and 12.71% and 2.33% from N2 to N3, respectively. It can be seen that increasing nitrogen fertilizer can significantly improve soil urease and sucrase activities, but the increase gradually decreases with increasing nitrogen application. Increasing nitrogen fertilizer application can reduce soil pH, but the treatments have no significant difference (*p* > 0.05).

### 3.5. Correlation Analysis between Various Indicators of Spring Maize

The yield of spring maize mainly comes from the photosynthetic products during the filling period [31], so the root and crown growth and photosynthetic gas exchange parameters during the grain filling period are selected as evaluation indicators. Figure 10 showed a significant positive correlation (*p* < 0.05) between yield and water consumption, transpiration rate, above-ground biomass, root biomass, root-to-shoot ratio, urease, and sucrase under degradable plastic film coverage. The yield under plastic film coverage was positively correlated with water consumption, above-ground biomass, root biomass, root-to-shoot ratio, urease, and sucrase (*p* < 0.05) and negatively correlated with pH.

### 3.6. Construction of a Comprehensive Growth Evaluation Model for Spring Maize

#### 3.6.1. Comprehensive Evaluation Hierarchy Model

They were using Yaaph software to establish a hierarchical model for the comprehensive evaluation of spring maize (Figure 11). The total growth index (C) target layer includes four criteria layers: yield and water use index (C1), photosynthetic index (C2), root and crown growth index (C3), and soil index (C4). The yield indicators include two indicator layers: yield (C11) and water consumption (C12). The photosynthetic indicators include three indicator layers: net photosynthetic rate (C21), transpiration rate (C22), and stomatal conductance (C23). The root cap growth indicators include three indicator layers: above-ground dry matter mass (C31), root mass (C32), and root cap ratio (C33). Soil quality indicators include three indicator layers: urease (C41), sucrase (C42), and pH (C43).

#### 3.6.2. Indicator Weights

##### AHP Method for Determining Indicator Weights

After establishing the hierarchical model, a judgment matrix is specified using a scale of 1–9. According to Figure 10, values are assigned to each indicator, and the consistency of the judgment matrix is checked. The judgment matrices for the comprehensive growth indicator (C), yield indicator (C1), root and crown growth indicator (C3), and soil indicator (C4) are as follows:

Degradable plastic film:$$C=\left[\begin{array}{cccc} 1.0000 & 2.0000 & 2.5000 & 3.0000\\ 0.5000 & 1.0000 & 0.5000 & 0.5000\\ 0.4000 & 2.0000 & 1.0000 & 2.0000\\ 0.3333 & 2.0000 & 0.5000 & 1.000 \end{array}\right]C1=\left[\begin{array}{cc} 1.0000 & 2.0000\\ 0.5000 & 1.0000 \end{array}\right]$$
$$C2=\left[\begin{array}{ccc} 1.0000 & 0.5000 & 1.0000\\ 2.0000 & 1.0000 & 2.0000\\ 1.0000 & 0.5000 & 1.0000 \end{array}\right]C3=\left[\begin{array}{ccc} 1.0000 & 2.0000 & 1.5000\\ 0.5000 & 1.0000 & 1.1000\\ 0.6667 & 0.9091 & 1.0000 \end{array}\right]$$
$$C4=\left[\begin{array}{ccc} 1.0000 & 0.5000 & 1.5000\\ 2.0000 & 1.0000 & 1.5000\\ 0.6667 & 0.6667 & 1.0000 \end{array}\right]$$

Ordinary plastic film:$$C=\left[\begin{array}{cccc} 1.0000 & 2.0000 & 2.5000 & 2.0000\\ 0.5000 & 1.0000 & 0.5000 & 1.5000\\ 0.4000 & 2.0000 & 1.0000 & 2.0000\\ 0.5000 & 0.6667 & 0.5000 & 1.000 \end{array}\right]C1=\left[\begin{array}{cc} 1.0000 & 1.5000\\ 0.6667 & 1.0000 \end{array}\right]$$
$$C2=\left[\begin{array}{ccc} 1.0000 & 2.0000 & 2.5000\\ 0.5000 & 1.0000 & 0.5000\\ 0.4000 & 2.0000 & 1.0000 \end{array}\right]C3=\left[\begin{array}{ccc} 1.0000 & 2.5000 & 1.5000\\ 0.4000 & 1.0000 & 0.5000\\ 0.6667 & 0.9091 & 1.0000 \end{array}\right]$$
$$C4=\left[\begin{array}{ccc} 1.0000 & 0.3333 & 0.5000\\ 3.0000 & 1.0000 & 2.0000\\ 2.0000 & 0.5000 & 1.0000 \end{array}\right]$$

The consistency test coefficients CR of the comprehensive growth index (C), yield index (C1), root and shoot growth index (C3), and soil index (C4) of the two plastic film mulchings were all less than 0.1, indicating that the consistency test results were good. The established judgment matrix was reliable and reasonable (Table 6, λmax is the maximum eigenvalue). The results showed that the weight of each index under degradable plastic film mulching was in the order of yield, water consumption, above-ground dry matter quality, sucrase, transpiration rate, root-shoot ratio, root quality, urease, pH, net photosynthetic rate, and stomatal conductance. The weight of each index under ordinary plastic film mulching was in the order of yield, water consumption, above-ground dry matter quality, net photosynthetic rate, root-shoot ratio, sucrase, stomatal conductance, root quality, pH, transpiration rate, and urease.

##### Entropy Weight Method for Determining Indicator Weights

The weights of various indicators of spring maize were calculated using Matlab programming, as shown in Table 7. According to the table, the consequences of multiple indicators under degradable plastic film cover, in descending order, were: pH, stomatal conductance, root-to-shoot ratio, root mass, net photosynthetic rate, urease, water consumption, above-ground dry matter mass, yield, sucrase, and transpiration rate. Under ordinary plastic film cover, the weights of various indicators were in descending order: pH, stomatal conductance, net photosynthetic rate, root mass, above-ground dry matter mass, urease, sucrase, and water consumption root-to-shoot ratio, yield, and transpiration rate.

##### Combination Weight Determination Based on the Game Theory

To avoid the influence of subjective factors on evaluation, an essential weight set formula was constructed based on two weighting values obtained from the AHP method and the entropy weighting method:$$w=\sum_{k=1}^{l}\alpha_{k}\times w_{k}^{T}(\alpha_{k}>0)$$
where α_k_, w_k_ are the weights obtained from the AHP method and the entropy weight method.

Calculate the weight set model based on game theory and derive the formula for the game model: $\mathrm{Min}=\left\|\sum_{j=1}^{i}a_{j}\times u_{i}^{T}-u_{i}^{T}\right\|(i=1,2)$. The normalized combination coefficients of the formula can be obtained using Matlab: a_1_ = 0.8507, a_2_ = 0.1493 (D); a_1_ = 0.7881, a_2_ = 0.2119 (P). Thus, the combined weight vector was obtained, and the final result is shown in Table 8. As shown in the table, the weights of various indicators under degradable plastic film cover in descending order were yield, water consumption, above-ground dry matter mass, sucrase, root-to-shoot ratio, root mass, transpiration rate, urease, pH, stomatal conductance, and net photosynthetic rate. Under ordinary plastic film cover, the weights of various indicators in descending order were yield, water consumption, above-ground dry matter mass, net photosynthetic rate, root-to-shoot ratio, sucrase, pH, stomatal conductance, root quality, transpiration rate, and urease.

#### 3.6.3. Comprehensive Growth Evaluation of Spring Maize Based on TOPSIS Method

Established a TOPSIS comprehensive evaluation model with combined weighting, normalize the decision matrix, established a weighted matrix, and calculated the ideal solution and fit degree C_i_ of the evaluation index. The calculation results were shown in Table 9. As shown in the table, DN3 treatment had the highest comprehensive index of adhesion (0.8522) for spring maize, followed by PN2 treatment (0.8435), and DN0 treatment had the lowest bonding (0.0194), indicated that poor comprehensive performance of spring maize.

## 4. Discussion

### 4.1. Effect of Film Mulching Combined with Nitrogen Fertilizer Application on Root and Shoot Growth

Crop root and shoot growth is more sensitive to nitrogen fertilizer. Increasing nitrogen fertilizer application can accelerate crop growth, root and shoot growth and development, and increase nitrogen uptake. However, excessive or insufficient nitrogen application can change crop growth morphology, affecting dry matter distribution and accumulation [32,33]. In the early stage of maize growth, degradable plastic film and ordinary plastic film coverage can form a “diaphragm effect” to significantly promote maize growth. In the later growth stage, degradable plastic film coverage degrades, which is beneficial for rainfall infiltration. In addition, the same amount of irrigation provides a good water and fertilizer environment for maize growth, with little impact on crop reproductive growth. Therefore, the effect of ordinary plastic film coverage and the application of nitrogen fertilizer with plastic film coverage on crop root and crown growth is consistent; under the same nitrogen application level, there was no significant difference in the root and crown growth of maize covered with degradable plastic film and ordinary plastic film, which was similar to the research conclusions of Huang et al. [34] and Wang et al. [21]. The root system is the main organ for crops to absorb nutrients. Increasing nitrogen fertilizer application can promote the growth of maize roots and increase root biomass, and the relationship between root biomass and nitrogen application is non-linear. When nitrogen application exceeds 320 kg·ha^−1^, it will inhibit root growth and development and reduce root biomass. This was consistent with the research conclusion of Qi et al. [35], which indicated that reasonable nitrogen fertilizer management measures can contribute to the formation of maize root morphology and increase root quality. The results of this study also indicated that there was a non-linear relationship between the accumulation of above-ground dry matter in maize and nitrogen application; that is, nitrogen application exceeding 320 kg·ha^−1^ will affect maize growth and reduce above-ground biomass, which was consistent with the research results of Li et al. [36]. Appropriate nitrogen fertilizer management measures can promote the development of maize roots, benefit the accumulation of above-ground biomass, form a reasonable root cap ratio, and lay the foundation for high crop yield.

### 4.2. Effect of Film Mulching Combined with Nitrogen Fertilizer Application on Photosynthetic Gas Exchange Characteristics

Photosynthesis is the process by which crops convert inorganic substances in the atmosphere, such as water and carbon dioxide, into organic matter and release oxygen. Crops automatically adapt to environmental changes and develop in a direction that is conducive to photosynthesis [37]. The future way to increase crop yield will mainly rely on the increase in photosynthetic conversion rate [38]. The results of this study indicated that increasing nitrogen fertilizer application can significantly enhance the photosynthetic capacity of maize leaves, and the photosynthetic gas exchange parameters (net photosynthetic rate, transpiration rate, and stomatal conductance of maize) showed an approximately linear relationship with increasing nitrogen application rate. At a nitrogen application level of 480 kg·ha^−1^, net photosynthetic rate, transpiration rate, and stomatal conductance were the highest, rising by 2.87~45.71%, 5.06~69.97%, and 12.70~71.24% under the degradable plastic film mulching, and 2.17~43.44%, 4.62~55.42%, and 9.79~73.64% under the ordinary mulching, respectively. This is because nitrogen can enhance the activity of mesophyll cells; increasing the SPAD value of leaves can improve photosynthesis [39], which was similar to the research conclusion of Gao et al. [40], and indicated that nitrogen fertilizer can improve the photosynthetic capacity of maize leaves. However, the degree of improvement varies due to factors such as crop variety, nitrogen fertilizer management measures, and the experimental environment. At the same time, this study also found that there was no significant difference in the photosynthetic gas exchange parameters between degradable plastic film-covered leaves and ordinary plastic film at the same nitrogen application level from the jointing stage to the grain-filling phase, indicating that the “diaphragm effect” formed by degradable plastic film and ordinary plastic film is the same [41], which can replace average plastic film to some extent.

### 4.3. Effect of Film Mulching Combined with Nitrogen Fertilizer Application on Maize Yield and Water and Nitrogen Use Efficiency

Reasonable nitrogen fertilizer management measures can promote root nutrient absorption, enhance crop assimilation, and increase yield. This study showed that the yield changed with nitrogen application rate in a quadratic parabolic relationship, and the yield-increasing effect slowed down with an increase in nitrogen application rate, which was in line with the diminishing returns effect. Moreover, excessive nitrogen application will reduce yield because it will affect crop nitrogen absorption efficiency, reduce nitrogen transport rate, and even affect root water absorption, resulting in a decreased yield [42,43]. Increasing the application of nitrogen fertilizer can enhance the water absorption capacity of the root system [44]. The results of this study indicated that increasing the application of nitrogen fertilizer can improve the water use efficiency and irrigation water use efficiency of maize. However, with the increase in nitrogen application level, the agronomic use efficiency and partial productivity of nitrogen fertilizer tended to decrease, which was consistent with the research conclusions of Li et al. [1]. Therefore, the appropriate amount of nitrogen fertilizer application provided a good soil environment for root growth, which improved crop yield and water use efficiency.

### 4.4. Effect of Film Mulching Combined with Nitrogen Fertilizer Application on Soil Enzyme Activity and pH

Soil enzyme activity, as an essential component of soil microbial activity and soil fertility, plays a critical catalytic role in soil nutrient cycling and energy conversion and can reflect the impact of fertilization on soil fertility and quality [45]. The results of this study indicated that the coverage area affected urease and sucrase activities. Under the same nitrogen application level, soil urease and sucrase activities under ordinary plastic film cover were higher than those under degradable plastic film, but there was no significant difference. This was similar to the research conclusions of Yang et al. [46] and Chen et al. [47]. Still, the reduction amplitude varies due to factors such as experimental materials, the experimental area environment, and field management measures. The application of nitrogen fertilizer can significantly increase the activities of urease and sucrase, which were due to the promotion of microbial activity by nitrogen fertilizer, changes in microbial composition, and thus affected soil enzyme activity [48], which was consistent with the research findings of Li et al. [48]. This study also found that increasing nitrogen fertilizer application can control soil salinity, reduce soil pH, and avoid soil salinization, which was consistent with the findings of Fudjoe et al. [49]. Increasing the application of nitrogen fertilizer can promote root development, enhance soil microbial activity, improve soil fertility, and reduce soil salinity, which is conducive to the sustainable development of agriculture.

## 5. Conclusions

Plastic film mulching and nitrogen fertilizer application are essential in agricultural production. The results of this study indicated that although the root and shoot growth, photosynthesis, and grain yield of spring maize under degradable plastic film mulching were lower than those under ordinary film mulching, there was no significant difference found. Nitrogen fertilizer was the main factor affecting spring maize growth and grain yield formation. When the nitrogen application rate approached 320 kg·ha^−1^, spring maize root growth and root biomass might be promoted, and there was no significant difference in net photosynthetic rate, transpiration rate, and stomatal conductance compared to the nitrogen application rate of 480 kg·ha^−1^. Under the nitrogen application level of 320 kg·ha^−1^, the yield of degradable plastic film and ordinary plastic film coverage increased by 3.74~42.50% and 2.05~40.02%, respectively, while spring maize had the highest water use efficiency and irrigation water use efficiency. However, nitrogen fertilizer’s agronomic utilization efficiency and partial productivity showed a decreasing trend. At the same time, there was no significant difference in soil enzyme activity between the nitrogen application level and 480 kg·ha^−1^. After conducting a comprehensive evaluation of the impact of plastic film mulching combined with nitrogen fertilizer on the growth of spring maize, using multiple indicators, it was found that the best overall growth of corn was achieved by using a nitrogen application rate of 320 kg·ha^−1^ with degradable plastic film mulching. Therefore, this strategy is optimal for plastic film mulching combined with nitrogen fertilizer application.

## Figures and Tables

**Figure 1 plants-13-01241-f001:**
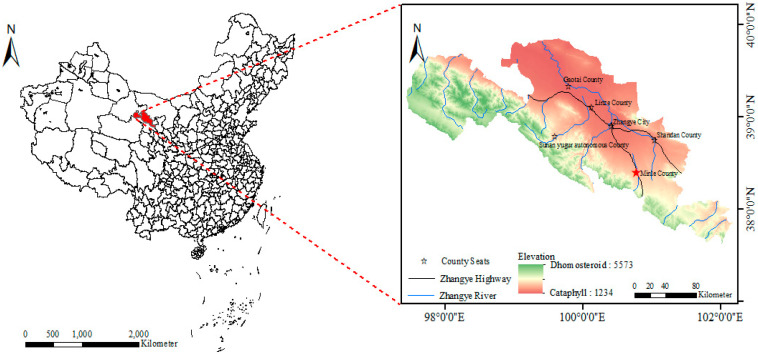
Location of the experimental site. Red star represents the city where the experimental site is located.

**Figure 2 plants-13-01241-f002:**
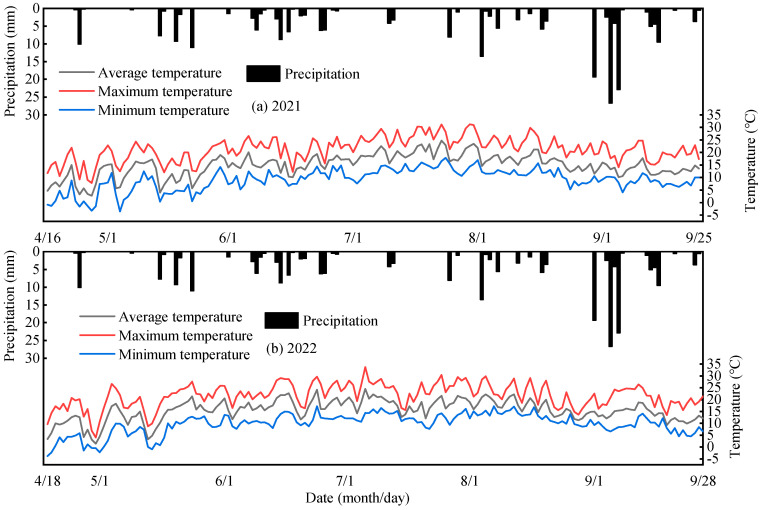
Precipitation and temperature during the spring maize growth period in 2021 (**a**) and 2022 (**b**).

**Figure 3 plants-13-01241-f003:**
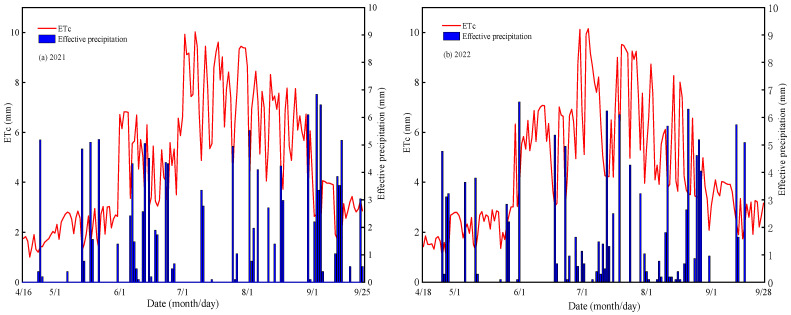
ETc and the effective precipitation during the spring maize growth period in 2021 (**a**) and 2022 (**b**).

**Figure 4 plants-13-01241-f004:**
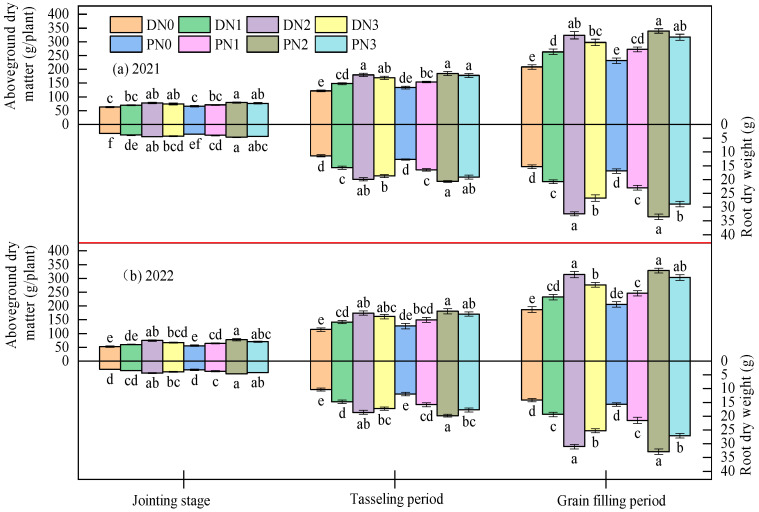
Effects of different mulching and nitrogen application on root and shoot growth of spring maize in 2021 (**a**) and 2022 (**b**). D represents degradable plastic film, P represents ordinary plastic film, and N0, N1, N2, and N3 represent 0, 160, 320, and 480 kg·ha^−1^ nitrogen fertilizer. The letters above the histogram indicate that there are significant differences among different treatments (*p* < 0.05). The data in the figure are the average of multiple repeated sets (n = 3). The bar above the bar graph represents the standard error.

**Figure 5 plants-13-01241-f005:**
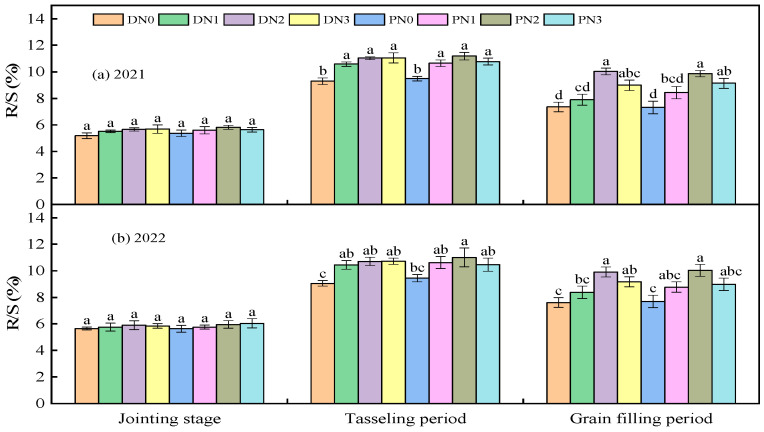
Effect of different film mulching and nitrogen application on spring maize root shoot ratio in 2021 (**a**) and 2022 (**b**). R/S represents root-to-shoot ratio, D represents degradable plastic film, P represents ordinary plastic film, and N0, N1, N2, and N3 represent 0, 160, 320, and 480 kg·ha^−1^ nitrogen fertilizer. The letters above the histogram indicate that there are significant differences among different treatments (*p* < 0.05). The data are the figure is the average of multiple repeated sets (n = 3). The bar above the bar graph represents the standard error.

**Figure 6 plants-13-01241-f006:**
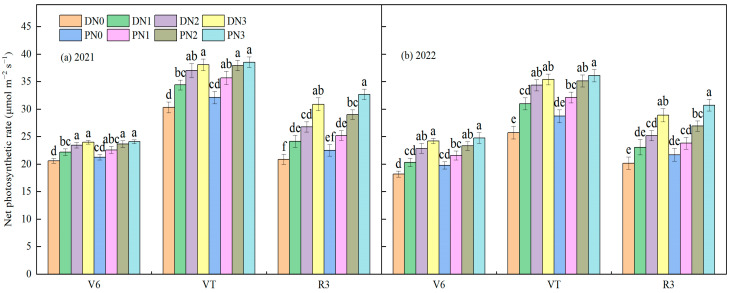
Effect of different film mulching combined with nitrogen fertilizer on the net photosynthetic rate of spring maize (2021 (**a**) and 2022 (**b**)). D represents degradable plastic film, P represents ordinary plastic film, and N0, N1, N2, and N3 represent 0, 160, 320, and 480 kg·ha^−1^ nitrogen fertilizer. The letters above the histogram indicate that there are significant differences among different treatments (*p* < 0.05). The data in the figure are the average of multiple repeated sets (n = 3). The bar above the bar graph represents the standard error.

**Figure 7 plants-13-01241-f007:**
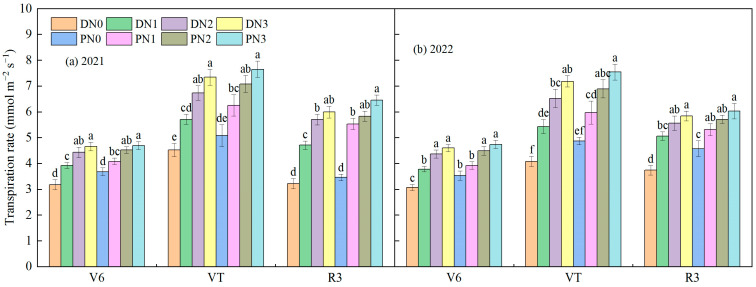
The effect of different coverage and nitrogen fertilizer applications on the spring maize transpiration rate (2021 (**a**) and 2022 (**b**)). D represents degradable plastic film, P represents ordinary plastic film, and N0, N1, N2, and N3 represent 0, 160, 320, and 480 kg·ha^−1^ nitrogen fertilizer. The letters above the histogram indicate that there are significant differences among different treatments (*p* < 0.05). The data in the figure are the average of multiple repeated sets (n = 3). The bar above the bar graph represents the standard error.

**Figure 8 plants-13-01241-f008:**
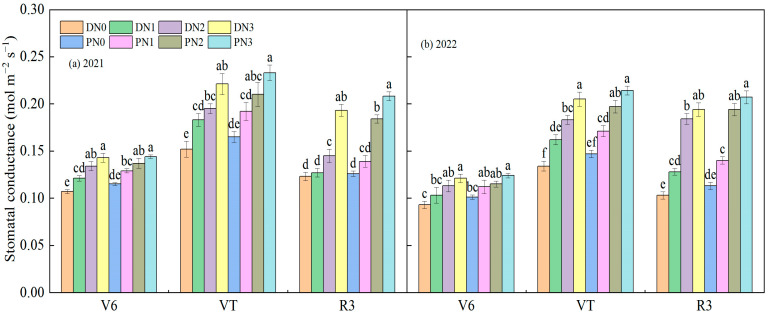
The effect of different coverage and nitrogen fertilizer application on the stomatal conductance of spring maize (2021 (**a**) and 2022 (**b**)). D represents degradable plastic film, P represents ordinary plastic film, and N0, N1, N2, and N3 represent 0, 160, 320, and 480 kg·ha^−1^ nitrogen fertilizer. The letters above the histogram indicate that there are significant differences among different treatments (*p* < 0.05). The data in the figure are the average of multiple repeated sets (n = 3). The bar above the bar graph represents the standard error.

**Figure 9 plants-13-01241-f009:**
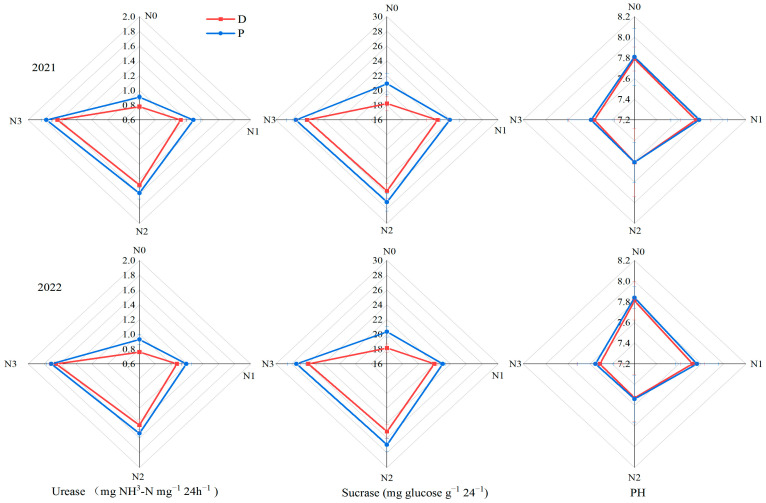
The effect of film mulching combined with nitrogen fertilizer on the soil quality of spring maize. D represents degradable plastic film, P represents ordinary plastic film, and N0, N1, N2, and N3 represent 0, 160, 320, and 480 kg·ha^−1^ nitrogen fertilizer.

**Figure 10 plants-13-01241-f010:**
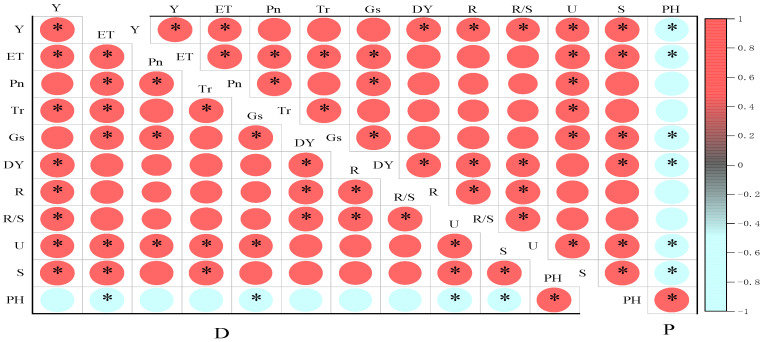
Correlation analysis between various indicators of spring maize under different treatments, * Significant difference at *p* < 0.05 level. D represents degradable plastic film, P represents ordinary plastic film, Y represents yield, ET represents crop evapotranspiration, Pn represents net photosynthetic rate, Tr represents transpiration rate, Gs represents stomatal conductance, DY represents above-ground biomass, R represents root biomass, R/S represents root to shoot ratio, U represents urease, and S represents sucrase.

**Figure 11 plants-13-01241-f011:**
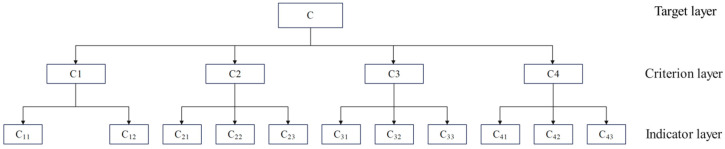
Comprehensive growth evaluation model diagram of spring maize.

**Table 1 plants-13-01241-t001:** Significance test on spring maize root and shoot growth at different growth stages; ns means no significant difference (*p* > 0.05); * means significant at *p* < 0.05 level; ** means significant at *p* < 0.01 level.

Year	F Fest	Jointing		Tasseling		Grain Filling	
		Above-Ground Dry Matter	Root Dry Matter	Above-Ground Dry Matter	Root Dry Matter	Above-Ground Dry Matter	Root Dry Matter
2021	F	1.16 ns	3.83 ns	4.35 ns	5.12 *	5.39 *	8.16 *
	N	12.44 **	29.36 **	43.94 **	102.72 **	45.27 **	139.07 **
	F×N	0.04 ns	0.16 ns	0.16 ns	0.27 ns	0.15 ns	0.19 ns
2022	F	4.09 ns	4.31 ns	2.38 ns	5.52 *	7.01 *	10.18 **
	N	29.27 **	27.05 **	18.33 **	58.71 **	63.02 **	157.72 **
	F×N	0.02 ns	0.06 ns	0.05 ns	0.26 ns	0.20 ns	0.04 ns
	Y	34.57 **	9.79 **	3.48 ns	12.39 **	16.10 **	8.94 **
	F	4.73 *	8.05 **	5.87 *	10.55 **	12.31 *	18.20 **
	N	39.39 **	54.90 **	51.40 **	151.20 **	106.92 **	295.61 **
	Y×F	0.38 ns	0.21 ns	0.03 ns	0.19 ns	0.03 ns	0.01 ns
	Y×N	1.52 ns	0.90 ns	0.03 ns	0.25 ns	0.66 ns	0.10 ns
	F×N	0.02 ns	0.01 ns	0.16 ns	0.52 ns	0.28 ns	0.16 ns
	Y×F×N	0.05 ns	0.19 ns	0.00 ns	0.01 ns	0.07 ns	0.08 ns

**Table 2 plants-13-01241-t002:** Significance test of spring maize root shoot ratio at different growth stages; ns means no significant difference (*p* > 0.05); * means significant at *p* < 0.05 level; ** means significant at *p* < 0.01 level.

Year	F Fest	Jointing Stage	Tasseling Period	Grain Filling Period
2021	F	0.41 ns	0.03 ns	0.22 ns
	N	1.77 ns	19.24 **	17.72 **
	F×N	0.12 ns	0.38 ns	0.32 ns
2022	F	0.11 ns	0.27 ns	0.13 ns
	N	0.60 ns	6.23 **	10.43 **
	F×N	0.07 ns	0.25 ns	0.17 ns
	Y	4.73 *	1.41 ns	0.86 ns
	F	0.45 ns	0.28 ns	0.33 ns
	N	2.03 ns	19.40 **	27.09 **
	Y×F	0.02 ns	0.13 ns	0.00 ns
	Y×N	0.13 ns	0.10 ns	0.25 ns
	F×N	0.01 ns	0.56 ns	0.35 ns
	Y×F×N	0.17 ns	0.01 ns	0.12 ns

**Table 3 plants-13-01241-t003:** Significance test of photosynthetic gas exchange parameters of spring maize at different growth stages. Pn represents the net photosynthetic rate, Tr represents the transpiration rate, and Gs represents stomatal conductance. ns means no significant difference (*p* > 0.05); * means significant at *p* < 0.05 level; ** means significant at *p* < 0.01 level.

Year	F Fest	Jointing Stage			Tasseling Period			Grain filling Period		
		Pn	Tr	Gs	Pn	Tr	Gs	Pn	Tr	Gs
2021	F	0.84 ns	3.02 ns	4.00 ns	2.10 ns	3.73 ns	3.88 ns	5.74 *	8.71 **	21.70 **
	N	15.14 **	24.54 **	33.63 **	17.82 **	26.54 **	21.23 **	38.33 **	81.79 **	86.84 **
	F×N	0.12 ns	0.89 ns	0.44 ns	0.13 ns	0.08 ns	0.04 ns	0.12 ns	1.19 ns	4.31 *
2022	F	3.16 ns	4.22 ns	2.24 ns	3.39 ns	5.83 *	8.07 *	3.42 ns	4.57 *	8.84 **
	N	19.89 **	34.39 **	8.29 **	25.51 **	34.77 **	56.66 **	23.08 **	21.97 **	136.09 **
	F×N	0.27 ns	0.61 ns	0.23 ns	0.50 ns	0.22 ns	0.11 ns	0.10 ns	0.92 ns	0.04 ns
	Y	7.13 *	1.15 ns	71.52 **	35.52 **	2.26 ns	21.89 **	7.42 *	1.15 ns	0.72 ns
	F	3.97 ns	7.14 *	5.56 *	5.42 *	9.36 **	10.16 **	8.85 **	12.50 **	28.98 **
	N	34.23 **	58.08 **	31.60 **	43.00 **	60.68 **	62.68 **	59.28 **	88.49 **	213.84 **
	Y×F	0.97 ns	0.02 ns	0.02 ns	0.08 ns	0.06 ns	0.02 ns	0.04 ns	0.07 ns	1.28 ns
	Y×N	2.68 ns	0.21 ns	0.88 ns	0.39 ns	0.07 ns	0.23 ns	0.28 ns	5.04 **	10.15 **
	F×N	0.38 ns	1.47 ns	0.44 ns	0.55 ns	0.26 ns	0.11 ns	0.21 ns	0.77 ns	2.03 ns
	Y×F×N	0.07 ns	0.05 ns	0.16 ns	0.09 ns	0.03 ns	0.01 ns	0.01 ns	1.30 ns	2.23 ns

**Table 4 plants-13-01241-t004:** Effects of different mulching and nitrogen application on spring maize yield and water and nitrogen use efficiency. The data in the table is the mean ± standard deviation, n = 3. Different letters after the same column of numbers indicate significant differences (*p* < 0.05); ns means no significant difference (*p* > 0.05); * means significant at *p* < 0.05 level; ** means significant at *p* < 0.01 level.

Year	Treatments	Yield (kg·ha^−1^)	ET (mm)	WUE (kg·m^−3^)	IWUE (kg·m^−3^)	NPF (kg·kg^−1^)	NFA (kg·kg^−1^)
2021	DN0	10,056.06 ± 723.38 d	786.20 ± 16.74 ab	1.28 ± 0.07 d	1.60 ± 0.12 d	--	--
	DN1	12,275.78 ± 497.49 bc	818.83 ± 14.01 ab	1.50 ± 0.05 bcd	1.96 ± 0.08 bc	76.72 ± 3.11 a	28.83 ± 3.03 a
	DN2	14,016.28 ± 542.33 ab	839.86 ± 15.91 a	1.67 ± 0.10 a	2.24 ± 0.11 ab	43.80 ± 2.04 b	12.38 ± 0.40 b
	DN3	13,680.4 ± 653.61 ab	844.57 ± 20.03 a	1.62 ± 0.03 abc	2.18 ± 0.08 ab	28.50 ± 1.13 c	7.55 ± 0.83 c
	PN0	11,014.06 ± 622.97 cd	770.25 ± 21.94 b	1.43 ± 0.10 cd	1.76 ± 0.10 cd	--	--
	PN1	12,973.96 ± 514.01 ab	793.97 ± 16.97 ab	1.63 ± 0.05 abc	2.07 ± 0.08 ab	81.09 ± 3.21 a	12.25 ± 1.28 b
	PN2	14,895.24 ± 586.73 a	827.76 ± 20.65 ab	1.80 ± 0.03 a	2.38 ± 0.10 a	46.55 ± 1.83 b	12.13 ± 0.23 b
	PN3	14,621.44 ± 607.53 a	835.34 ± 14.52 a	1.75 ± 0.09 a	2.33 ± 0.10 a	30.46 ± 1.27 c	7.52 ± 0.24 c
F fest	F	4.23 ns	1.52 ns	6.38 *	4.25 ns	2.71 ns	24.17 **
	N	18.00 **	5.05 *	9.52 **	17.95 **	251.76 **	44.21 **
	F×N	0.02 ns	0.07 ns	0.01 ns	0.02 ns	0.15 ns	22.97 **
2022	DN0	9446.10 ± 446.82 d	746.38 ± 11.50 cd	1.26 ± 0.04 d	1.55 ± 0.07 d	--	--
	DN1	11,207.88 ± 488.95 cd	765.96 ± 16.27 bcd	1.46 ± 0.07 cd	1.84 ± 0.08 cd	70.05 ± 3.06 a	25.21 ± 0.88 a
	DN2	13,773.96 ± 766.25 ab	801.97 ± 14.61 ab	1.72 ± 0.12 ab	2.26 ± 0.13 ab	43.04 ± 2.39 b	13.52 ± 1.31 bc
	DN3	13,107.30 ± 518.96 abc	813.98 ± 15.09 a	1.61 ± 0.03 abc	2.15 ± 0.09 abc	27.31 ± 1.08 c	7.63 ± 0.18 d
	PN0	10,028.44 ± 895.66 d	727.34 ± 10.15 d	1.38 ± 0.11 cd	1.65 ± 0.15 d	--	--
	PN1	12,017.44 ± 626.16 bc	758.35 ± 13.62 bcd	1.59 ± 0.10 bc	1.97 ± 0.10 bc	75.11 ± 3.91 a	12.43 ± 2.92 bcd
	PN2	14,567.82 ± 498.79 a	784.20 ± 11.10 abc	1.86 ± 0.05 a	2.39 ± 0.08 a	45.52 ± 1.56 b	14.19 ± 1.25 b
	PN3	14,248.88 ± 573.33 a	795.25 ± 16.05 ab	1.79 ± 0.07 ab	2.34 ± 0.09 a	29.69 ± 1.19 c	8.79 ± 1.16 cd
F fest	F	3.61 ns	2.65 ns	6.64 *	3.75 ns	2.78 ns	8.60 *
	N	21.63 **	9.98 **	14.52 **	21.65 **	169.04 **	24.24 **
	F×N	0.07 ns	0.08 ns	0.09 ns	0.07 ns	0.20 ns	13.45 **
	Y	4.45 *	25.85 **	0.00 ns	0.83 ns	4.09 ns	0.02 ns
	F	7.81 **	3.88 ns	13.29 **	7.97 **	5.48 *	11.74 **
	N	39.48 **	13.72 **	24.50 **	39.48 **	412.74 **	25.52 **
	Y×F	0.00 ns	0.00 ns	0.00 ns	0.00 ns	0.01 ns	0.53 ns
	Y×N	0.29 ns	0.06 ns	0.35 ns	0.38 ns	1.77 ns	0.54 ns
	F×N	0.05 ns	0.01 ns	0.03 ns	0.05 ns	0.34 ns	13.78 **
	Y×F×N	0.04 ns	0.14 ns	0.07 ns	0.05 ns	0.01 ns	0.12 ns

**Table 5 plants-13-01241-t005:** Significance test of soil enzyme activity and pH. ns means no significant difference (*p* > 0.05); * means significant at *p* < 0.05 level; ** means significant at *p* < 0.01 level.

Year	F Fest	Urease	Sucrase	pH
2021	F	9.66 **	4.33 ns	0.02 ns
	N	76.05 **	15.13 **	0.48 ns
	F×N	0.06 ns	0.12 ns	0.002 ns
2022	F	6.16 *	8.61 *	0.06 ns
	N	64.21 **	37.80 **	1.32 ns
	F×N	0.24 ns	0.20 ns	0.003 ns
	Y	1.21 ns	0.45 ns	0.08 ns
	F	15.49 **	11.18 **	0.06 ns
	N	139.26 **	43.66 **	1.57 ns
	Y×F	0.10 ns	0.02 ns	0.00 ns
	Y×N	0.24 ns	0.04 ns	0.04 ns
	F×N	0.14 ns	0.25 ns	0.01 ns
	Y×F×N	0.17 ns	0.04 ns	0.00 ns

**Table 6 plants-13-01241-t006:** Weight calculation results of AHP Analytic Hierarchy Process.

	Degradable Plastic Film			Ordinary Plastic Film		
	Local Weights	Final Weight	Consistency Check Parameters	Local Weights	Final Weight	Consistency Check Parameters
Target layer C	0.4435	0.4435	C_R_ = 0.0664 < 0.1 λmax = 4.1774	0.4115	0.4115	C_R_ = 0.0477 < 0.1 λmax = 4.1274
	0.1360	0.1360		0.1781	0.1781	
	0.2493	0.2493		0.2604	0.2604	
	0.1713	0.1713		0.1460	0.1460	
Criterion layer C1	0.6667	0.2957	C_R_ = 0.0000 < 0.1 λmax = 2.0000	0.6000	0.2469	C_R_ = 0.0000 < 0.1 λmax = 2.0000
	0.3333	0.1478		0.4000	0.1646	
Criterion layer C2	0.2500	0.0340	C_R_ = 0.0000 < 0.1 λmax = 3.0000	0.5232	0.0932	C_R_ = 0.0904 < 0.1 λmax = 3.0940
	0.5000	0.0680		0.1928	0.0343	
	0.2500	0.0340		0.2840	0.0506	
Criterion layer C3	0.4641	0.1157	C_R_ = 0.0157 < 0.1 λmax = 3.0163	0.4797	0.1249	C_R_ = 0.0036 < 0.1 λmax = 3.0037
	0.2636	0.0657		0.1805	0.0470	
	0.2723	0.0679		0.3398	0.0885	
Criterion layer C4	0.2918	0.0500	C_R_ = 0.0516 < 0.1 λmax = 3.0536	0.1634	0.0239	C_R_ = 0.0088 < 0.1 λmax = 3.0092
	0.4632	0.0793		0.5396	0.0788	
	0.2451	0.0420		0.2970	0.0434	

**Table 7 plants-13-01241-t007:** Single index weights of spring maize calculated based on Entropy Weight Method.

Treatments	Index	C_11_	C_12_	C_21_	C_22_	C_23_	C_31_	C_32_	C_33_	C_41_	C_42_	C_43_
D	Weight	0.0797	0.0837	0.0922	0.0754	0.1102	0.0834	0.0923	0.0983	0.0864	0.0785	0.1199
P		0.0787	0.0817	0.0980	0.0717	0.0999	0.0858	0.0862	0.0802	0.0834	0.0822	0.1522

**Table 8 plants-13-01241-t008:** Determination of Single Index Weights for spring maize Based on Game Theory through Combination Weighting.

Treatments	Index	C_11_	C_12_	C_21_	C_22_	C_23_	C_31_	C_32_	C_33_	C_41_	C_42_	C_43_
D	Weight	0.2634	0.1382	0.0427	0.0691	0.0454	0.1109	0.0697	0.0724	0.0554	0.0792	0.0536
P		0.2113	0.1470	0.0942	0.0422	0.0610	0.1166	0.0553	0.0867	0.0365	0.0795	0.0665

**Table 9 plants-13-01241-t009:** Comprehensive indicators and ranking of spring maize based on TOPSIS method. S^+^ represents the ideal solution, S^−^ represents the inverse perfect solution, D^+^ represents the distance between each processing and the ideal solution, and D^−^ represents the distance between each processing and the inverse perfect solution.

Treatments	C_11_	C_12_	C_21_	C_22_	C_23_	C_31_	C_32_	C_33_	C_41_	C_42_	C_43_	D^+^	D^−^	C_i_	Sorted
DN0	0.3963	0.4774	0.4066	0.3435	0.3685	0.3703	0.3066	0.4288	0.2998	0.3930	0.5091	0.2023	0.0040	0.0194	8
DN1	0.4772	0.4936	0.4678	0.4821	0.4175	0.4647	0.4181	0.4666	0.4263	0.4792	0.5048	0.1245	0.0825	0.3985	5
DN2	0.5647	0.5114	0.5147	0.5557	0.5381	0.5979	0.6614	0.5719	0.5664	0.5485	0.4941	0.0322	0.1855	0.8522	1
DN3	0.5443	0.5166	0.5924	0.5838	0.6327	0.5380	0.5420	0.5209	0.6384	0.5613	0.4918	0.0413	0.1755	0.8097	4
S^+^	0.5647	0.5166	0.5924	0.5838	0.6327	0.5979	0.6614	0.5719	0.6384	0.5613	0.5091				
S^−^	0.3963	0.4774	0.4066	0.3435	0.3685	0.3703	0.3066	0.4288	0.2998	0.3930	0.4918				
PN0	0.3999	0.4757	0.4121	0.3700	0.3570	0.3846	0.3169	0.4255	0.3284	0.4162	0.5091	0.1781	0.0043	0.0235	7
PN1	0.4749	0.4931	0.4568	0.4995	0.4165	0.4567	0.4325	0.4875	0.4408	0.4731	0.5055	0.1130	0.0689	0.3789	6
PN2	0.5599	0.5121	0.5216	0.5318	0.5623	0.5880	0.6452	0.5640	0.5586	0.5452	0.4928	0.0300	0.1616	0.8435	2
PN3	0.5487	0.5180	0.5911	0.5751	0.6188	0.5457	0.5443	0.5131	0.6211	0.5530	0.4925	0.0323	0.1584	0.8308	3
S^+^	0.5599	0.5180	0.5911	0.5751	0.6188	0.5880	0.6452	0.5640	0.6211	0.5530	0.5091				
S^−^	0.3999	0.4757	0.4121	0.3700	0.3570	0.3846	0.3169	0.4255	0.3284	0.4162	0.4925				

## Data Availability

The datasets used and/or analyzed during the current study are available from the corresponding author upon reasonable request and the approval of the data owner.

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
