# Peer review of "Optimizing the Mulching Pattern and Nitrogen Application Rate to Improve Maize Photosynthetic Capacity, Yield, and Nitrogen Fertilizer Utilization Efficiency"

_plants, 2024, doi:10.3390/plants13091241_

Round 1
Reviewer 1 Report
Comments and Suggestions for Authors
This paper describes studies with biodegradable mulches compared with other treatments on maize growth and development. It is not completely novel but relates to particular environments. There is very detailed study of a range of physiological and growth parameters and detailed analysis .
Some of the detail and analysis is excessive and could be placed in an annex.
This paper needs careful editing as many results and comments are written as statements not reports of results and in present tense and not past tense .
The use of singular and plural verbs needs correcting.
Authors should describe When and how fertilizer was applied . Was it applied after the mulch was placed.
Is it likely that the degradable plastic allowed improved translocation of N into the soil ?
line 56. This discussion of the environmental effects of plastic mulches needs clarification:
Do mulches increase or decrease emissions of CO2, CH4 and NH4 ?
Do mulches improve sequestering of CO2 and NH4 ?
There are detailed edits up to line 379 and then general criticism of the writing style. ( see attached file

see above . Needs rewiting to improve english and presentation
Author Response
Dear Reviewers:
Thank you for your comments concerning our manuscript entitled “Can degradable plastic film replace the ordinary under various nitrogen applications in Spring maize production?” (Manuscript ID: plants-2925974). The comments were all valuable and helpful in revising and improving our manuscript, in addition to contextualizing the significance of our research. The reviewer comments are laid out below in italicized font and specific concerns have been numbered. Revised components of the manuscript are indicated by the " Track Changes" function within the uploaded revised file.
Responses to Reviewer 1 Comments
Comment 1: The rainfall in 2021 and 2022 was 244.7 mm and 237.4 mm, respectively (Line149).
Response 1: Thank you for pointing out this problem in manuscript.In order to improve the rationality and completeness of the paper, We have made changes in the text, and the rainfall here is the effective rainfall shown in Figure 3. The rainfall in line 106 represents the total rainfall, which is shown in Figure 2. Please review.
Comment 2: Line 46 suggest use critical instead of sever.
Response 2: Thank you for pointing out this problem in manuscript.In order to improve the rationality and completeness of the paper, We have replaced the words and highlighted them in red font in the text. Please review.
Comment 3: Line62 Which is degraded very slowly in soil.
Response 3: Thank you for pointing out this problem in manuscript.In order to improve the rationality and completeness of the paper, We have replaced it and highlighted it in red font in the text for your review.
Comment 4: Line 85 suggest use critical instead of sever.
Response 4: Thank you for pointing out this problem in manuscript.In order to improve the rationality and completeness of the paper, We have replaced the words and highlighted them in red font in the text. Please review.
Comment 5: Line 90 to maintian high yields.
Response 5: Thank you for pointing out this problem in manuscript.In order to improve the rationality and completeness of the paper, We have replaced it and highlighted it in red font in the text for your review.
Comment 6: Line 91. rewritw as :" excessive application of nitorgen fertilizer does not ....".
Response 6: Thank you for pointing out this problem in manuscript.In order to improve the rationality and completeness of the paper, We have replaced it and highlighted it in red font in the text for your review.
Comment 7: Line 210 replace "while" with "and".
Response 7: Thank you for pointing out this problem in manuscript.In order to improve the rationality and completeness of the paper, We have replaced the words and highlighted them in red font in the text. Please review.
Comment 8: Line 220 replace "with" with "and ".
Response 8: Thank you for pointing out this problem in manuscript.In order to improve the rationality and completeness of the paper, We have replaced the words and highlighted them in red font in the text. Please review.
Comment 9: Line 221 replace "with" with "and ".
Response 9: Thank you for pointing out this problem in manuscript.In order to improve the rationality and completeness of the paper, We have replaced the words and highlighted them in red font in the text. Please review.
Comment 10: Line 329 : Insert "This showed that ...".
Response 10: Thank you for pointing out this problem in manuscript.In order to improve the rationality and completeness of the paper, We have replaced the words and highlighted them in red font in the text. Please review.
Comment 11: line 56. This discussion of the environmental effects of plastic mulches needs clarification: Do mulches increase or decrease emissions of CO2, CH4 and NH4 ? Do mulches improve sequestering of CO2 and NH4 ?
Response 11: Thank you for pointing out this problem in manuscript.In order to improve the rationality and completeness of the paper, We have confirmed that the specific description in the literature is to increase the emission pressure of N2O and CH4 gases, without describing CO2 gas. Rewrite the summary and highlight it in red font in the text for your review.

Reviewer 2 Report
Comments and Suggestions for Authors
The manuscript entitled “Can degradable plastic film replace the ordinary under various nitrogen applications in Spring maize production?”, concerns a two-year field experiment aiming at evaluating the effects of a biodegradable mulching film combined with different N fertilization levels on maize. The subject of the manuscript falls with the general scope of the Journal and provides some interesting data. However, it shows many gaps and weaknesses that need to be addressed.
GENERAL COMMENTS:
· The manuscript is not formatted according to the journal guidelines. Please use the template and follow the specific instructions
· Overall, the level of English language is low and it should be enhanced with the help of a mother-tongue proof-reader
· The entire manuscript is too long in my opinion. There several redundant information and other ones (more important) not properly addressed. I suggest reducing by 20-30% the manuscript length (abstract, introduction and results)
· The quality of writing, in general, is low. Please see specific comments
· The number of references is too high (61), with some ones not relevant and other ones lacking. Please adjust the list of references
· The number of tables and figures is too high. Some of them could be combined for an easier understanding for readers. Other ones, could be moved to supplementary materials
SPECIFIC COMMENTS:
· The abstract is too long in my opinion. I suggest to summarise it and to highlight better the main findings
· L14-15: “and biodegradable plastic film mulching is acknowledged as the best alternative to ordinary plastic films”
· L16: “… coupled with nitrogen application…”
· Keywords are not appropriate since they are composite words too long
· L44-48: this section is quite confusing. Please rephrase it
· L49-61: I think this section is too long and could be reduced. The main subject of this manuscript are biodegradable plastic mulch films, so please focus more on them
· On the contrary, the background about biodegradable plastic mulch films should be introduced better. I strongly recommend this recent review paper that could be useful both for authors and readers and can allow reducing the number of references: “Soil Bioplastic Mulches for Agroecosystem Sustainability: A Comprehensive Review”, https://doi.org/10.3390/agriculture13010197
· L76-83: the behaviour of biodegradable plastic mulch films is not always like this. Here, you are referring to specific works in specific pedo-climatic conditions and with a specific film type. However, as reported in the text they seem a general trend. Therefore, please be specific and introduce the context.
· L84-89: This section reports well-known information available in all agronomic books. Please delete it
· L99-102: is this an hypothesis or is it corroborated by previous findings?
· Fig. 2: please specify in the caption what the sub-figures up and down refer to. Moreover, specify also with the x-axe refer. I suggest also moving the precipitation in the lower part of the graph, together with air temperatures
· L132-135: why did you not apply a border between plots? Especially, between nitrogen levels
· L165: please add the company and the country in brackets
· L168-174: anything is reported about soil sampling for analyses. How much soil samples? At which depth? How were they collected?
· Anything is reported about statistical analysis. Looking at results, I suppose you apply a two-way analysis of variance considering N fertilization level and mulching film type as fixed factors, and the year as random factor. But this should be added into the text.
· In addition, were the ANOVA basic assumptions (homoscedasticity and normality) verified before the analysis? If yes, how? Which test did you apply for means comparisons?
· Figures 4-8: please spell-out the acronyms and treatments in each caption
Comments on the Quality of English LanguageThe level of English level, although I am not a native English, is low. The manuscript will benefit if reviewed by a mother-tongue proof-reader.
Author Response
Dear Reviewers:
Thank you for your comments concerning our manuscript entitled “Can degradable plastic film replace the ordinary under various nitrogen applications in Spring maize production?” (Manuscript ID: plants-2925974). The comments were all valuable and helpful in revising and improving our manuscript, in addition to contextualizing the significance of our research. The reviewer comments are laid out below in italicized font and specific concerns have been numbered. Revised components of the manuscript are indicated by the " Track Changes" function within the uploaded revised file.
Responses to Reviewer 2 Comments
Comment 1: The manuscript is not formatted according to the journal guidelines. Please use the template and follow the specific instructions.
Response 1: Thank you for pointing out this problem in manuscript.In order to improve the rationality and completeness of the paper, We have made corrections to the parts of the manuscript that do not meet the requirements according to the journal format, and marked them in the text for your review.
Comment 2: Overall, the level of English language is low and it should be enhanced with the help of a mother-tongue proof-reader.
Response 2: Thank you for pointing out this problem in manuscript.In order to improve the rationality and completeness of the paper, We have rewritten and reviewed this manuscript, and have corrected any grammar issues in the manuscript. They have been highlighted in red font in the text for your review.
Comment 3: The number of references is too high (61), with some ones not relevant and other ones lacking. Please adjust the list of references.
Response 3: Thank you for pointing out this problem in manuscript.In order to improve the rationality and completeness of the paper, Based on your revision suggestions, we have removed some content from the introduction and reduced the number of references to 53, and marked them in the text. Please review.
Comment 4: The abstract is too long in my opinion. I suggest to summarise it and to highlight better the main findings.
Response 4: Thank you for pointing out this problem in manuscript.In order to improve the rationality and completeness of the paper, We have summarized the manuscript abstract and highlighted it in red font in the text for your review.
Comment 5: L16: “… coupled with nitrogen application…”.
Response 5: Thank you for pointing out this problem in manuscript.In order to improve the rationality and completeness of the paper, We have made modifications and highlighted them in red font in the text for your review.
Comment 6: “and biodegradable plastic film mulching is acknowledged as the best alternative to ordinary plastic films”.
Response 6: Thank you for pointing out this problem in manuscript.In order to improve the rationality and completeness of the paper, We have deleted it and marked it in the text for your review.
Comment 7: Keywords are not appropriate since they are composite words too long.
Response 7: Thank you for pointing out this problem in manuscript.In order to improve the rationality and completeness of the paper, We have made modifications to the keywords in the manuscript and highlighted them in red font in the text. Please review.
Comment 8: L44-48: this section is quite confusing. Please rephrase it.
Response 8: Thank you for pointing out this problem in manuscript.In order to improve the rationality and completeness of the paper, We have summarized lines 44-48 and rewritten them, highlighting them in red font in the text. Please review.
Comment 9: L49-61: I think this section is too long and could be reduced. The main subject of this manuscript are biodegradable plastic mulch films, so please focus more on them.
Response 9: Thank you for pointing out this problem in manuscript.In order to improve the rationality and completeness of the paper, We have re summarized lines 49-61, deleted some content, and marked them in the manuscript for your review.
Comment 10: L49-61: On the contrary, the background about biodegradable plastic mulch films should be introduced better. I strongly recommend this recent review paper that could be useful both for authors and readers and can allow reducing the number of references: “Soil Bioplastic Mulches for Agroecosystem Sustainability: A Comprehensive Review”,
Response 10: Thank you for pointing out this problem in manuscript.In order to improve the rationality and completeness of the paper, Thank you very much for your feedback. We have carefully read this manuscript and it does provide guidance for our future writing. At the same time, we have also made modifications to our introduction based on the application of degradation membranes in this manuscript. Please review.
Comment 11: L84-89: This section reports well-known information available in all agronomic books. Please delete it.
Response 11: Thank you for pointing out this problem in manuscript.In order to improve the rationality and completeness of the paper, We have deleted lines 84-89 and marked them in the text for your review.
Comment 12: L99-102: is this an hypothesis or is it corroborated by previous findings?
Response 12: Thank you for pointing out this problem in manuscript.In order to improve the rationality and completeness of the paper, As confirmed by previous research results, lines 99 to 102 are referenced in the manuscript as 1. Ridge-furrow muching combined with appropriate nitrogen rate for enhancing photosynthetic efficiency, yield and water use efficiency of summer size in a semi arid region of China; 38. Is it biodegradable film an alternative to polyethylene plastic film for improving maize production in rainbow agricultural areas? - Evidence from field experiments and 50. The application of water worthy plastic film and biodegradable film as alternative to polyethylene film in cryos on the Loss Plateau has been confirmed.
Comment 13: Fig. 2: please specify in the caption what the sub-figures up and down refer to. Moreover, specify also with the x-axe refer. I suggest also moving the precipitation in the lower part of the graph, together with air temperatures.
Response 13: Thank you for pointing out this problem in manuscript.In order to improve the rationality and completeness of the paper, The X-axis represents time, and the black bar chart in the upper part of Figure 2 represents the rainfall during the maize growth period, which is also explained in the legend. The lower part of the graph represents the temperature during the growth period, which is the highest temperature, lowest temperature, and average temperature. We separate the rainfall and temperature in the graph to avoid duplication with published paper images. Please understand.
Comment 14: L132-135: why did you not apply a border between plots? Especially, between nitrogen levels.
Response 14: Thank you for pointing out this problem in manuscript.In order to improve the rationality and completeness of the paper, We have added community boundaries in the article and marked them in red font. We will pay attention to such issues in the future paper writing process. Thank you very much for your feedback, please review.
Comment 15: L165: please add the company and the country in brackets.
Response 15: Thank you for pointing out this problem in manuscript.In order to improve the rationality and completeness of the paper, We have indicated the biodegradable plastic film production company in our writing and highlighted it in red font in the text. At the same time, the production country has also been added and marked in red font. Please review.
Comment 16: L168-174: anything is reported about soil sampling for analyses. How much soil samples? At which depth? How were they collected?
Response 16: Thank you for pointing out this problem in manuscript.In order to improve the rationality and completeness of the paper, We have added soil sampling depth and sampling method in section 2.3.3 for soil sample determination and analysis, and highlighted them in red font in the text. Please review.
Comment 17: Anything is reported about statistical analysis. Looking at results, I suppose you apply a two-way analysis of variance considering N fertilization level and mulching film type as fixed factors, and the year as random factor. But this should be added into the text.
Response 17: Thank you for pointing out this problem in manuscript.In order to improve the rationality and completeness of the paper, In the writing of the entire article, we have always applied the two factor difference analysis, which analyzes nitrogen levels and plastic film types, without considering the analysis of inter annual differences in data. At the same time, based on your feedback, "The number of tables and figures is too high. Some of them could be combined for an annual understanding for readers. Other ones, could be moved to supplementary materials." We did not include inter annual data difference analysis.
Comment 18: In addition, were the ANOVA basic assumptions (homoscedasticity and normality) verified before the analysis? If yes, how? Which test did you apply for means comparisons?
Response 18: Thank you for pointing out this problem in manuscript.In order to improve the rationality and completeness of the paper, In the analysis of variance, the hypothesis of homogeneity of variance test is met, that is, the result of Levin's equivalence test in the analysis of variance is greater than 0.05 (based on the significance corresponding to the mean), indicating homogeneity of variance, and the data can be further analyzed. To visually represent the homogeneity test results, we selected a set of data analysis results to be presented in the form of images. Please review.
Comment 19: Figures 4-8: please spell-out the acronyms and treatments in each caption.
Response 19: Thank you for pointing out this problem in manuscript.In order to improve the rationality and completeness of the paper, We have added the corresponding processing for each test code in Figures 4-8, and provided additional explanations for the missing parts, which are highlighted in red font in the text. Please review.
Round 2
Reviewer 2 Report
Comments and Suggestions for Authors
The authors addressed some comments and suggestions from the previous review, while other ones were not considered. I therefore reiterate some comments with the aim of improving the quality of the manuscript:
- the level of Enlish language is still quite low, with several grammar mistakes and wrong sentences. I encourage the authors to review the paper with the help of a mother-tongue proof-reader;
- the manuscript is still not formatted according to journal guidelines. Please download the template from the journal website and follows it with accuracy;
- I don't really like interrogative titles, but this is a personal opinion. I suggest an affermative title;
- please divide the fifth keyword into two different keywords;
- I suggest adding some economic data about maize harvest area and production in China;
- In my opinion, the number of references is still high. Some of them could be deleted, especially in the introduction, and changed with a single comprehensive paper. See for instance L48-52 (MS with track changes), which references may be changed with the following article: https://doi.org/10.3390/agriculture13010197
- paragraph 2.4: I suggest naming it "Statistical analysis". Please incorporate in the text the answers to my previous comments N. 17 and 18. "Anything is reported about statistical analysis. Looking at results, I suppose you apply a two-way analysis of variance considering N fertilization level and mulching film type as fixed factors, and the year as random factor. But this should be added into the text.". "In addition, were the ANOVA basic assumptions (homoscedasticity and normality) verified before the analysis? If yes, how? Which test did you apply for means comparisons?".
- Figures 4-8 and Tables 2-5: please include in the caption the post-hoc test used for mean multiple comparisons
- Maybe some Figures (e.g. 10, 11) and Tables (e.g. 6-9) can be moved to supplementary materials?
Comments on the Quality of English Languagethe level of Enlish language is still quite low, with several grammar mistakes and wrong sentences. I encourage the authors to review the paper with the help of a mother-tongue proof-reader
Author Response
Dear Reviewers:
Thank you for your comments concerning our manuscript entitled “Can degradable plastic film replace the ordinary under various nitrogen applications in Spring maize production?” (Manuscript ID: plants-2925974). The comments were all valuable and helpful in revising and improving our manuscript, in addition to contextualizing the significance of our research. The reviewer comments are laid out below in italicized font and specific concerns have been numbered. Revised components of the manuscript are indicated by the " Track Changes" function within the uploaded revised file.
Responses to Reviewer 2 Comments
Comment 1: the level of Enlish language is still quite low, with several grammar mistakes and wrong sentences. I encourage the authors to review the paper with the help of a mother-tongue proof-reader;
Response 1: Thank you for pointing out this problem in manuscript.In order to improve the rationality and completeness of the paper, We have rewritten and reviewed this manuscript, and have corrected any grammar issues in the manuscript. They have been highlighted in red font in the text for your review.
Comment 2: the manuscript is still not formatted according to journal guidelines. Please download the template from the journal website and follows it with accuracy.
Response 2: Thank you for pointing out this problem in manuscript.In order to improve the rationality and completeness of the paper, We have made corrections to the parts of the manuscript that do not meet the requirements according to the journal format, and marked them in the text for your review.
Comment 3: I don't really like interrogative titles, but this is a personal opinion. I suggest an affermative title.
Response 3: Thank you for pointing out this problem in manuscript.In order to improve the rationality and completeness of the paper, Based on your feedback and the content of the manuscript, we have made corrections to the title and highlighted it in red font in the text. Please review.
Comment 4: please divide the fifth keyword into two different keywords.
Response 4: Thank you for pointing out this problem in manuscript.In order to improve the rationality and completeness of the paper,We have made changes to the fifth keyword, please review it.
Comment 5: I suggest adding some economic data about maize harvest area and production in China.
Response 5: Thank you for pointing out this problem in manuscript.In order to improve the rationality and completeness of the paper,We have supplemented the situation of corn planting area in China in the manuscript and highlighted it in red font in the text. However, we have not found any data on the economic benefits of corn, only the total agricultural output value data in China, so we have not supplemented it. Please review.
Comment 6: In my opinion, the number of references is still high. Some of them could be deleted, especially in the introduction, and changed with a single comprehensive paper. See for instance L48-52 (MS with track changes), which references may be changed with the following article.
Response 6: Thank you for pointing out this problem in manuscript.In order to improve the rationality and completeness of the paper,We have reviewed the literature you recommended and found that it meets the citation requirements of our manuscript. Therefore, we have reduced the references from lines 48 to 52 to one. At the same time, the number of references in the entire manuscript has been reduced to 49.
Comment 7: paragraph 2.4: I suggest naming it "Statistical analysis".
Response 7: Thank you for pointing out this problem in manuscript.In order to improve the rationality and completeness of the paper,We have replaced it, please review it.
Comment 8: Figures 4-8 and Tables 2-5: please include in the caption the post-hoc test used for mean multiple comparisons.
Response 8: Thank you for pointing out this problem in manuscript.In order to improve the rationality and completeness of the paper,We have added your comments in the manuscript and highlighted them in red font for your review.
Comment 9: Maybe some Figures (e.g. 10, 11) and Tables (e.g. 6-9) can be moved to supplementary materials?
Response 9: Thank you for pointing out this problem in manuscript.In order to improve the rationality and completeness of the paper,We have carefully considered and would appreciate it if you could include Figures 10, 11, and Tables 6-9 in the supplementary materials for your review.
Comment 10: Anything is reported about statistical analysis. Looking at results, I suppose you apply a two-way analysis of variance considering N fertilization level and mulching film type as fixed factors, and the year as random factor. But this should be added into the text.
Response 10: Thank you for pointing out this problem in manuscript.In order to improve the rationality and completeness of the paper, According to your feedback, we have supplemented the data in Tables 1 to 5 for your review.
Comment 11: In addition, were the ANOVA basic assumptions (homoscedasticity and normality) verified before the analysis? If yes, how? Which test did you apply for means comparisons?
Response 11: Thank you for pointing out this problem in manuscript.In order to improve the rationality and completeness of the paper,Before data analysis, we conducted normality tests on the data and found that the data follows or approximates a normal distribution, and the data has homogeneous variance and significant differences between the data. The above data analysis was conducted using SPSS 24.0 software. The data analysis method was carried out according to the application of SPSS 25.0 in agricultural experimental statistical analysis edited by Zhou Xinbin (Chemical Industry Press). At the same time, we used a certain set of data for analysis and placed the analysis images below for your review.
